# Competitive capabilities of higher education institutions from their Employees' perspectives: A case study of King Khalid University

**Boshra Ismael Ahmed Arnout**[1,2]*, **Thabet Saeed AlQahtani**[3], **Hessah A. L. Melweth**[3]

**1** Department of Psychology, King Khalid University, Abha, Saudi Arabia, **2** Department of Psychology, Zagazig University, Zagazig, Egypt, **3** Faculty of Education, Department of Learning and Instructre, King Khalid University, Abha, Saudi Arabia

* prof.arnout74@gmail.com, beahmad@kku.edu.sa

**Data Availability Statement:** All data are fully available without restriction within the paper.

**Funding:** The author(s) received no specific funding for this work.

## Abstract

This study aimed to uncover the competitive advantages of King Khalid University (KKU) as a higher educational institution and identify the strategies needed to strengthen its competitive stance through a qualitative case study approach. Data were collected via detailed interviews with 30 university staff, comprising 19 faculty members and 11 academic leaders. Following this, the data were qualitatively analyzed using MAXQDA 2022 software. The results showed that KKU has 30 sub-competitive strengths, including work ethics, future vision, academic excellence, creativity, teamwork, respect for intellectual property, continuous customer-focused improvement, a positive workplace environment, organizational trust, and the ability to attract international students. Additionally, the study identified 8 challenges hindering KKU's advancement in global university rankings, spanning academic, human, and administrative areas. To improve its standing in international rankings, thematic analysis revealed 11 strategies to enhance KKU's competitiveness. These include aligning academic programs with job market demands, enhancing research facilities, boosting funding for academic and research endeavors, fostering international academic and scientific partnerships, and upgrading the technological infrastructure for academics and administration. The analysis underscores the need for KKU to adopt a comprehensive suite of academic, human, and administrative strategies to bolster its competitive position. This is crucial for KKU's rise in global university rankings and its alignment with the National Vision 2030, aiming to place over five Saudi universities among the top 100 or 200 globally.

## Introduction and theoretical literature

Higher education is now influenced by market standards such as profitability and quality, making competitiveness a key criterion for evaluating the excellence of institutions, including universities. This competitive environment is shaped by councils, bodies, policies, strategies,

**Competing interests:** The authors have declared that no competing interests exist.

and indicators. Internationalization has required universities to reassess their philosophies, policies, and all factors affecting their competitive capabilities. A prestigious university consistently ranks within the top 100 or 200 in international rankings, relies on a single funding source, and enjoys the highest level of academic freedom to develop its programs and conduct training and research in an environment that protects intellectual property rights. In this era, competitiveness presents a significant challenge for universities, forcing them to reevaluate their organizational structures and production capacities, reorganize their units, and optimize their resources to improve and sustain their competitive position and sustainability [1,2].

## Higher education institutions and competitiveness

Universities are indisputably the cornerstone of societal progress and evolution, driving society toward a brighter future by addressing its challenges and fostering growth and prosperity [1–5]. In the current era of globalization and openness, universities' role surpasses just education. They are crucial in enhancing society by developing graduates with 21st-century skills through outstanding academic programs offered by skilled faculty. This leads to the production of graduates who are not only competitive in the global job market but also endowed with knowledge, skills, responsibility, professional ethics, and a dedication to responsible citizenship. Such graduates play a key role in fulfilling national aspirations and achieving sustainable development goals [6–8].

Undoubtedly, when a university possesses competitive capabilities that ensure the production of globally competent graduates proficient in performing efficiently across various work environments, it sets itself up for success in international competitions and secures a high position in the International Competitiveness Report. This success is achieved by adhering to specific principles and standards. As a result, countries around the world are improving their educational and research infrastructures, technologies, and pedagogical methods, highlighting the importance of competitiveness. This is reflected in the significant quantitative and qualitative expansion of universities in the Kingdom, in line with the national vision for 2030. This expansion is accompanied by a move towards the privatization of higher education, propelled by increasing student numbers and demand [9–11].

Previously, the pursuit of competitive advantage was not considered essential within university settings. However, the introduction of international ranking systems and the criteria for obtaining quality and accreditation certifications have led universities to prioritize developing competitive capabilities that enable them to compete for prominence at local, regional, or international levels [12,13].

Since its inception in 2003, the Academic Ranking of World Universities, published by Shanghai's Jiao Tong University, has attracted global attention for its annual ranking of elite universities. The release of these rankings and the classification of universities on an international level is eagerly awaited each year. This anticipation is not limited to countries with prestigious universities but also extends to developing nations, increasingly focused on such rankings. Saudi Arabian universities, in particular, are deeply interested as they heavily invest in higher education and aim to see their investments reflected in the quality of their educational outcomes and their global academic standing. In an era characterized by rapid internationalization and the blurring of traditional boundaries, competitiveness has become crucial for societies, individuals, and institutions. To encourage this competitiveness, various policies, procedures, and benchmarks have been implemented [3–7].

Undoubtedly, the shift towards a knowledge-based society has significantly influenced higher education institutions, which are now required to adapt to the growing student body and the advancements in information and communication technologies. As a result, these

institutions find themselves obligated to develop strategies, policies, and procedures that foster excellence in every facet of their operations and decision-making processes, to maintain competitive advantages at local, regional, and international levels. Achieving this involves capitalizing on their distinct capabilities and characteristics that allow them to stand out in providing services to clients, thereby establishing themselves as frontrunners among their counterparts. This indicates a path toward sustained prosperity and resilience [5,10,11,14,15].

## The concept of university competitiveness

Competitiveness in higher education signifies a university's ability to provide educational, research, and community services effectively and of superior quality. This influences the institution's reputation, its graduates' marketability, and its faculty members' scholarly integrity and productivity [2,6,12]. Also, it includes the university's capability to maintain educational excellence, improve internal and external efficiency, meet demand, and enhance performance and outcomes, thus fulfilling local, international, and service objectives and attaining higher international rankings [11,14].

Furthermore, university competitiveness stems from internal factors such as resources, human skills, technology, organization, and financial assets, enabling the institution to provide outstanding educational and research services and to compete effectively on both local and international levels in education, scientific research, and community service. It also involves the university's ability to leverage its strengths and opportunities to offset weaknesses and reduce the impact of threats that may affect its outputs [4,14,16,17].

The World Economic Forum recognizes competitive capacity building as a complex process shaped by 12 essential factors, which together create the composite index for evaluating competitiveness. These factors fall into three main categories: (1) Basic requirements, covering institutions, infrastructure, macroeconomic stability, health, and primary education; (2) Efficiency enhancers, which include higher education and training, goods market efficiency, labor market efficiency, financial market development, technological readiness, and domestic market size; (3) Evolution and innovation factors, involving business structure evolution and innovation capacity. The Forum assesses the competitiveness index of 140 countries using available data and releases the findings in its yearly International Competitiveness Report. In its 2019 edition, the Kingdom was ranked 36th, whereas the 2023 IMD report placed the Kingdom of Saudi Arabia at 17th, according to the National Competitiveness Center [3,10,11,18].

Additionally, competitiveness is in line with the National Vision 2030's objective to rank at least five Saudi universities among the top 200 globally by 2030. This ambition is in harmony with the Kingdom's aspirations in higher education to compete for leadership, nurture a knowledge-based society, and meet the needs of economic and social progress. This transformation elevates Saudi universities from a national to a global level in education, research, and scientific publication. Furthermore, it complements the Ministry of Education's endeavors to foster a knowledge society and lead national innovation projects [10,11]. Also, corresponds with the Kingdom's political, economic, and technological objectives and its strategy for economic and human development that satisfies the aspirations and requirements of Saudi society. Competitiveness is essential for Saudi universities including King Khalid University, to sustain and invigorate their vitality, aid in formulating a new vision for their goals, execute competitive strategies, cultivate talent, prepare the job market with proficient and capable graduates, provide superior education, research, and community services, and highlight creativity and innovation to overcome both local and international competitors [6,10,14,15,19].

Achieving competitive advantage is crucial for any institution striving for excellence and success in today's competitive work environment. Given the role of universities in community

development, examining the competitive advantage of universities is significantly important. King Khalid University, a premier institution in Saudi Arabia, is committed to enhancing the quality of education and providing academic services of the highest standards. To achieve this, the university must continuously study and analyze its competitive advantage. Therefore, investigating King Khalid University's competitive advantage is necessary to help it in achieving excellence and success by evaluating its strengths and weaknesses, identifying areas where it excels over its competitors in the higher education market, and strengthening these aspects to maintain its distinction. Conversely, recognizing weaknesses and addressing them can increase its competitiveness. Furthermore, exploring the competitive advantage can aid King Khalid University in identifying opportunities for development and expansion. By understanding its market position, and assessing the needs of students and the community, the university can find new ways to enhance its services and attract more students. Additionally, exploring the competitive advantage can contribute to building King Khalid University's reputation and improving its market position. When the university stands out and delivers services that surpass those of its competitors, it will gain respect and appreciation from students, parents, and employers.

## Current study objectives

Numerous studies [10,11,13,20] have demonstrated that universities in the Kingdom are encountering a wide range of challenges that hinder their ability to achieve the goals of Saudi Vision 2030 and to improve their positions in global university rankings. Thus, this study is one of the first to explore King Khalid University's competitive advantage from the perspective of its employees, uncovering the strategies necessary for the university to secure a prominent spot in the global top 100 university rankings, to help King Khalid University achieve the 2030 nation view.

Research by Supe et al. [2] underscored the importance of identifying the factors influencing the competitiveness of higher education institutions. Therefore, this study aimed to assess the current capabilities of KKU and to identify the strategies necessary to enhance its competitive edge and climb in international rankings, leveraging insights from its administrators and faculty members through a qualitative case study methodology.

Accordingly, this study aims to address the following questions:

1. What are the KKU's current competitive capabilities?

2. What challenges does KKU face in achieving Advanced rank in International Ratings for the Best Universities?

3. How can KKU develop its competitive capabilities?

## 3. Methodology

### Method

This study utilized a qualitative case study methodology to explore KKU's competitive strengths through comprehensive in-depth interviews with participants, following their informed consent. Debout [21] defined a qualitative case study as a research method to explore a complex phenomenon by identifying diverse factors that interact with each other. The case is a real-life situation.

The in-depth interviews were carried out individually to accommodate the diverse work schedules of the participants. The researchers designed a protocol for the in-depth interviews,

comprising a set of broad, open-ended questions aligned with the research goals. These questions were designed to elicit insights into the participants' perspectives on King Khalid University's competitive advantages as a higher education institution, the challenges it faces, and the strategies needed to overcome these challenges to improve its competitive position and achieve a distinguished status in global university rankings. Questions included: What are your views on KKU's competitive advantages? What challenges does KKU encounter in its quest for higher positions in international university rankings? How can KKU bolster its global competitiveness? What strategies are crucial for reaching this goal?

The protocol was tested to ensure the questions' clarity and relevance to the study's aims and objectives. A total of thirty-eight individual interviews were conducted from August 20th to October 31st, 2023, each lasting between 45 and 60 minutes. Data collection continued until reaching a saturation point, confirmed when no new information emerged in the last four interviews. Additionally, field notes were recorded during the interviews with study participants.

## Participants

King Khalid University boasts a total of 3,407 faculty members. A purposive sample of academic leaders and faculty staff members at King Khalid University was selected to gather detailed insights into KKU's competitive strengths. In line with the objectives of the qualitative case study, the study's snowball sample included 30 purposively chosen individuals who serve as academic leaders and faculty members at KKU. These individuals possess extensive experience and uniformity in the work environment within the same institution (Table 1). They received an email invitation containing a study summary, its objectives, importance, methodology, and a request for their participation. Before conducting in-depth interviews, we confirmed the interviews' location and timing, obtained informed consent from the participants, and ensured the confidentiality of their responses by clarifying the study's purpose. Since this study is not experimental, it has been deemed exempt from IRB review

**Table 1. Sample demographic characteristics.**

| Demographic characteristics | N% |
|---|---|
| **Participants** | **30** |
| **Gender** | |
| Male | 16 (53.33%) |
| Female | 14 (46.67%) |
| **Role Job** | |
| Academic leaders | 11 (36.67%) |
| Faculty members | 19 (63.33% |
| **Years of Experience** | |
| 5–10 years | 11 (36.67%) |
| 11–15 years | 14 (46.67%) |
| Over 15 years | 5 (16.66%) |

Table 1 presents the demographic characteristics of the 30 participants employed at KKU. This cohort includes 14 females (46.67%) and 16 males (53.33%), comprising 11 academic leaders (36.67%) and 19 faculty members (63.33%). Their ages vary between 42 and 53 years, with an average age of 46.38 (± 5.18 years). Their tenure at the university ranges from 5 to over 15 years: 11 participants have between 5 to 10 years of experience (36.67%), 14 possess 11 to 15 years (46.67%), and 5 have more than 15 years (16.66%).

(No.23-049). Written informed consent was obtained from all participants involved in the study.

## Data analysis

The data gathered through detailed interviews with participants was analyzed. The researchers reviewed theoretical literature to understand the competitive capabilities of higher education institutions. In the process of examining the data, new theoretical concepts were identified. They conducted a thematic analysis focusing on the participants' perceptions, which were derived from interview transcripts, as well as their field notes and memos. Interview transcripts were copied, and the researchers then undertook repeated and thorough readings to grasp the deeper meanings before employing MAXQDA software to analyze and code qualitative data, thereby refining the answers to the research questions [22].

## Credibility, reliability, and transferability

Credibility in this study was established through participants' consistent responses. Researchers implemented various strategies to ensure data credibility, including meticulous re-reading and transcription of raw data from participant interviews, and individual analysis of each interview, followed by a comprehensive analysis to synthesize the data. Additionally, ensuring saturation was confirmed by selecting suitable participants. Reliability was ascertained through triangulation and extensive data collection from individual interviews by several researchers, along with independent coding of data by two experts in qualitative thematic analysis. Regarding transferability, extensive data were compiled to allow readers to evaluate the applicability of the study's findings to other contexts [23,24].

## Results

### Study question one: What are your perceptions of the current competitive capabilities of KKU?

To address this question, the researchers utilized an in-depth interview tool, enhancing it with extra questions pertinent to the primary inquiry. They performed a thematic analysis and coded the data using MAXQDA software. The findings are presented in Table 2, and Fig 1.

The thematic analysis of participant responses to the first question, as presented in Table 2 and Fig 1, reveals that KKU possesses 30 sub-competitive capabilities. These encompass work ethics, a vision for the future, academic excellence, innovation, and teamwork. Also included are respect for intellectual property, a commitment to continuous improvement for customer satisfaction, a positive organizational climate, and organizational trust. The analysis underscores business incubators, collaborations with the public and private sectors, marketing channels for university products through the shopping unit, a competitive message, clear objectives, and initiatives. It further highlights a dynamic strategy, primary and secondary specializations, the identification of strengths and weaknesses, and the anticipation of future labor market demands. Additional capabilities consist of an efficient administrative staff, advanced technology services, well-equipped classrooms and labs, technological administrative communications, online courses, a well-stocked central library, updated programs aligned with job market needs, an optimal learning environment, comprehensive study plans and courses, flexible and equitable admission policies, and the enrollment of international students.

**Table 2. Results of coding participants' responses to the first question using MAXQDA software.**

| Codes | Frequency | Sub-codes | Frequency |
|---|---|---|---|
| **Declared specific work values** | 28 | Work ethics | 8 |
| | | Foresight | 5 |
| | | Academic excellence | 4 |
| | | Innovation | 4 |
| | | Teamwork | 4 |
| | | Respect for intellectual property | 3 |
| **Comprehensive quality** | 24 | Continuous development to satisfy the customer | 9 |
| | | Positive organizational climate | 9 |
| | | Organizational trust | 6 |
| **Marketing university products locally and internationally** | 21 | Business incubators and partnerships with the public and private sectors | 12 |
| | | Marketing outlets for university products through the shopping unit | 9 |
| **Flexible strategy** | 18 | Competitive message, goals, and initiatives | 11 |
| | | Adaptable strategy | 7 |
| **Strategic management** | 16 | Primary and sub-specializations | 9 |
| | | Identifying strengths, weaknesses, opportunities, and threats | 4 |
| | | Measuring Future Labor Market Demand | 3 |
| **Academic and administrative staff** | 15 | Academically distinguished faculty members | 6 |
| | | Efficient administrative staff | 4 |
| | | Human resources management | 3 |
| | | Professional development programs | 2 |
| **Outstanding infrastructure and facilities** | 14 | Advanced technological services | 4 |
| | | Equipped classrooms and laboratories | 3 |
| | | Technological administrative communications | 3 |
| | | Electronic curricula | 2 |
| | | Equipped central library | 2 |
| **High-quality educational programs** | 12 | Developed programs that keep pace with the labor market | 3 |
| | | Suitable learning environment | 3 |
| | | Integrated study plans and curricula | 2 |
| | | Flexible and fair admission policies | 2 |
| | | Admission of international students | 2 |

## Study question two: What challenges are faced by the university of King Khalid University to achieve advanced rank in international ratings for the Best Universities?

To address research question two, researchers conducted in-depth interviews with participants. The collected data were subsequently analyzed using MAXQDA software. The results of the thematic analysis for this question are presented in Table 3, Fig 2.

Table 3 and Fig 2 reveal eight significant challenges impeding KKU's progress in global university rankings. These challenges encompass academic, human, and administrative barriers, including program categorization, graduate recognition by professional bodies, publications with international classifications, recruitment and employment challenges, resistance to change, a deteriorating faculty-to-student ratio, bureaucratic processes, and inadequate financial support for departments and programs.

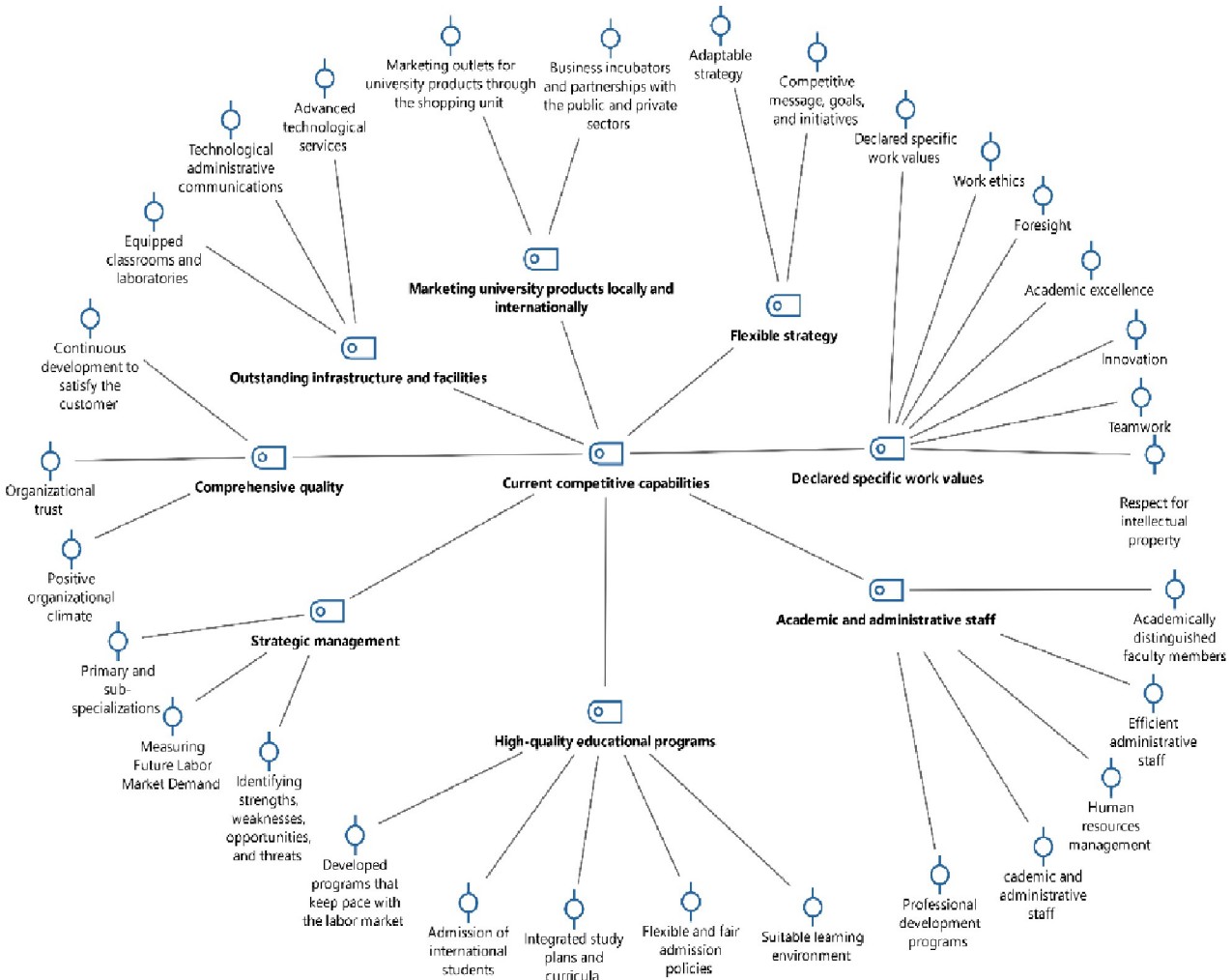

**Fig 1. Code-Subcode-Segment model of participants' responses to the first question via MAXQDA software.**

## Study question three: How does KKU develop its competitiveness?

To answer the third question of this study, the researchers utilized an interview tool to involve participants with sub-questions relevant to this inquiry. The collected data were analyzed thematically and then coded using the MAXQDA software. Table 4 and Fig 3 display the results of the analysis derived from participants' responses.

**Table 3. Results of coding participants' responses to the second question using MAXQDA software.**

| Codes | Frequency | Sub-codes | Frequency |
|---|---|---|---|
| **Academic challenges** | 23 | Program classification | 10 |
| | | Classification of graduates from specialty and professional bodies | 7 |
| | | International scientific publication | 6 |
| **Human challenges** | 20 | Difficulty of appointment and recruitment | 9 |
| | | Change resistance | 7 |
| | | Decreasing the number of teaching configurations for students in some disciplines | 4 |
| **Administrative challenges** | 18 | Administrative routines | 12 |
| | | Vulnerability of financial support for colleges and programs | 6 |

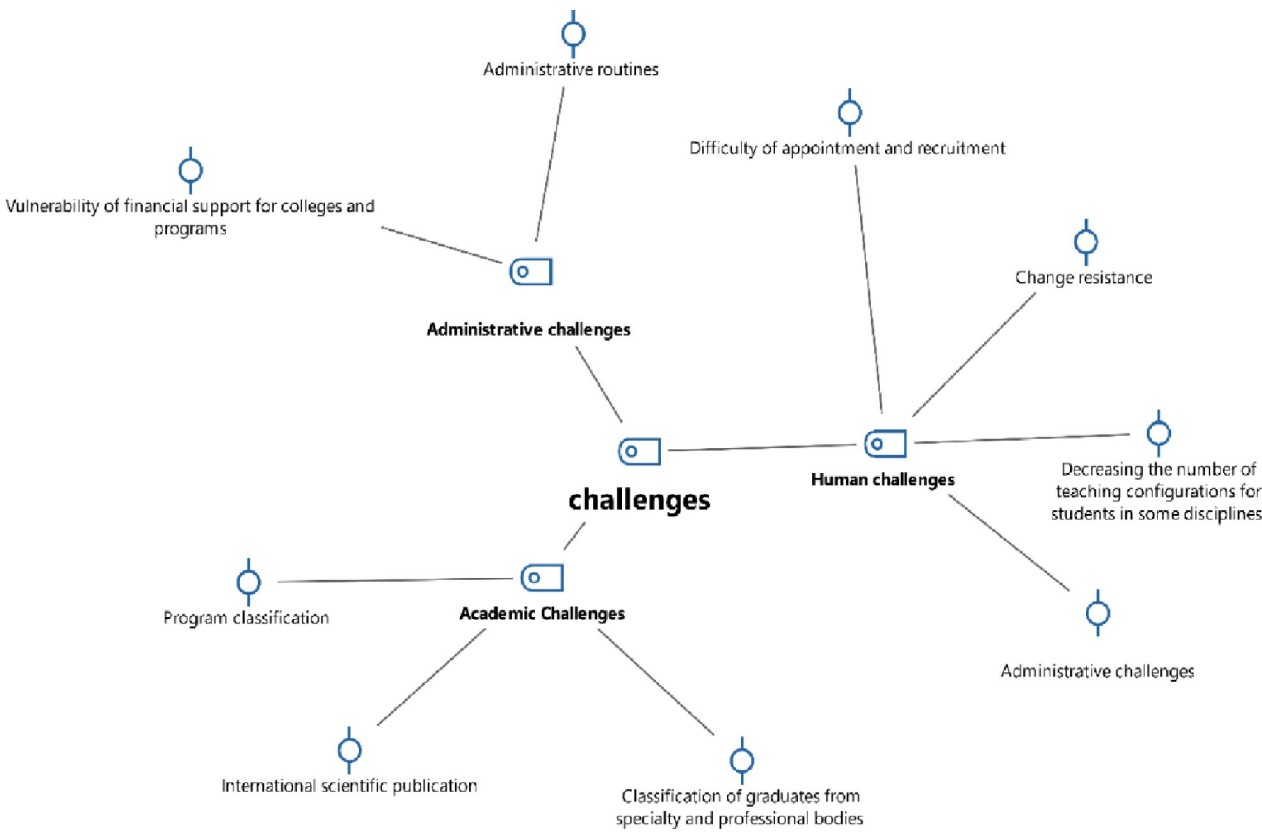

**Fig 2. Code-Subcode-Segment model of participants' responses to the second question via MAXQDA software.**

The thematic analysis of responses to the third question, as illustrated in Table 4 and Fig 3, indicates that elevating KKU's position in global rankings necessitates the identification of 11 strategies. These strategies encompass aligning academic programs with job market demands, introducing new academic offerings, attracting elite students, establishing specialized centers, expanding research facilities, enhancing funding for academic programs and research, fortifying connections with international universities, mandating training for faculty and staff,

**Table 4. Results of coding participants' responses to the third question using MAXQDA software.**

| Codes | Frequency | Sub-codes | Frequency |
|---|---|---|---|
| **Academic competitive** | 28 | Raising the level of compatibility of academic programs with the needs of the labor market | 8 |
| | | Introducing new academic programs | 5 |
| | | Attracting outstanding students | 4 |
| | | Building specialized centers | 3 |
| | | Expansion of research centers | 3 |
| | | Increasing the capabilities allocated to study programs and scientific research | 3 |
| | | Increasing scientific and academic ties with scientific universities | 2 |
| **Mechanisms of Human Competitiveness** | 23 | Mandatory training for faculty and staff | 12 |
| | | Attracting distinguished faculty members | 11 |
| **Managerial competitive mechanisms** | 22 | Improving the flexibility of systems and procedures | 18 |
| | | Raising the efficiency of the university's academic and administrative technological system | 2 |

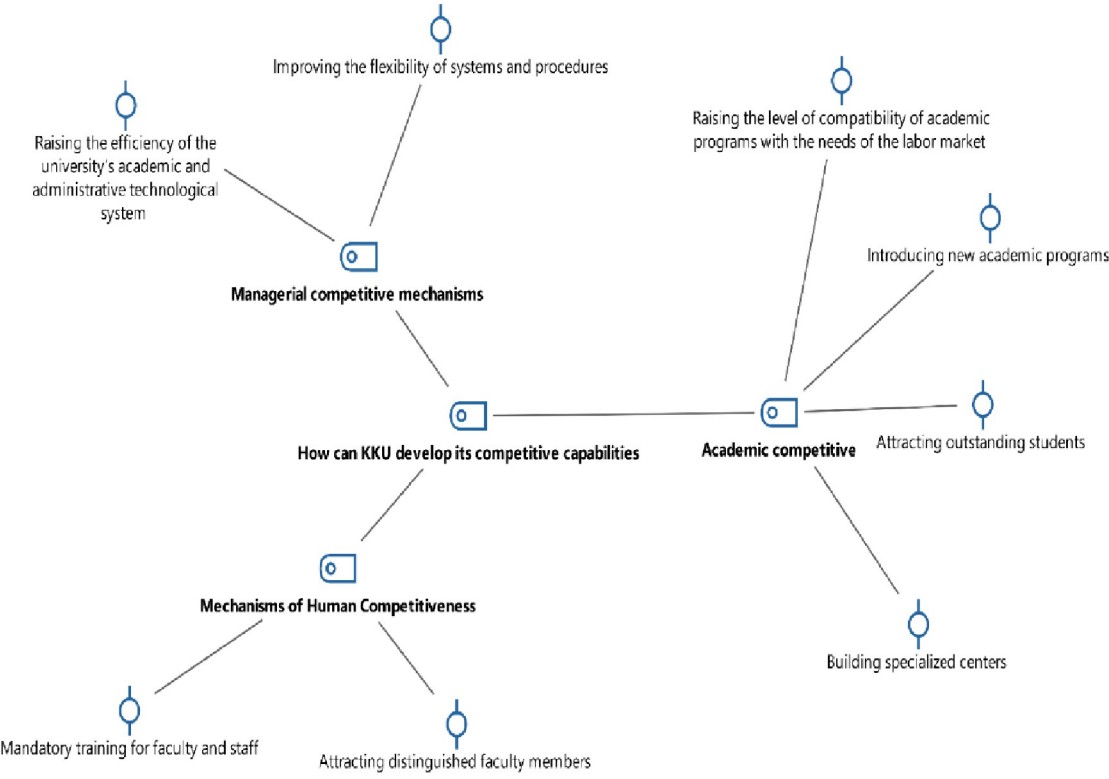

**Fig 3. Code-Subcode-Segment model of participants' responses to the third question via MAXQDA software.**

augmenting system and procedural flexibility, and refining the efficiency of academic and administrative technology systems.

## Discussion

The findings showed that King Khalid University boasts 30 competitive strengths, yet faces 8 challenges that impede its advancement in global university rankings. To elevate King Khalid University's standing in these rankings, qualitative data analysis identified 11 strategies to enhance its competitiveness.

### The current competitive capabilities of KKU

The findings of the current study, presented in Table 2, indicate that King Khalid University (KKU) possesses unique and clear work values among its workforce, as emphasized by its leadership. Remarkably, work ethics were mentioned 8 times. **Participant 1 observed, *"KKU's employees adhere to a core value of work ethics, encompassing integrity, confidentiality, and fulfilling client needs across all academic and administrative sectors." Similarly, Participant 2 remarked, "The university's staff is dedicated to executing their duties and serving stakeholders with a commitment to work ethics, ensuring prompt and satisfactory service."***

The university's leadership consistently demonstrates a forward-thinking approach. Participant 5 observed, ***"University employees follow well-defined plans."*** Participant 7 added, "***The university's strategic plans are specifically designed to meet future labor market demands."*** Furthermore, the commitment to academic excellence guides the performance of all university staff, a principle highlighted in four responses. Participant 9 stated, ***"Each university employee***

*follows a performance charter that details the skills and knowledge required for their academic and administrative roles*," while Participant 11 mentioned, *"I annually prepare my performance charter, which my direct supervisor then uses to evaluate my work proficiency*." Additionally, innovation is a fundamental value at KKU, also referenced four times. Participant 12 commented, *"University staff excel in unconventional work aimed at achieving excellence, innovation, and creativity,"* and Participant 13 noted, *"KKU's leadership encourages an innovative work environment, in both academic activities like teaching and research and administrative tasks*." This aligns with Curran's [25] study, which highlighted innovation as a crucial element in gaining competitive capabilities, especially in scientific research. In the same context, Chen et al. [26] revealed that universities enhance economic growth and boost employment opportunities by innovating through graduate training and offering career guidance.

KKU strongly values teamwork, as highlighted by four references. Participant 14 observed, *"University staff across colleges and administrative units collaborate seamlessly as one team,"* and Participant 15 added, *"All university staff unite to fulfill the university's strategic objectives as a cohesive system*." Additionally, respect for intellectual property stands as a fundamental value, mentioned three times. For example, Participant 18 remarked, "*We commit to the ethical guidelines of scientific research and honor the intellectual property rights of others in our research endeavors*." Furthermore, Participant 19 stated, "*The university's research and scientific contributions are characterized by innovation and a commitment to intellectual property and ethical standards*."

These findings are consistent with the values Al-Abbad [11] identified as essential for competitiveness capabilities, grounded in scientific planning to align with the evolving programs, studies, and research in response to changes in the local, regional, and international labor markets. The university also prioritizes academic excellence, fostering sound scientific reasoning, creativity, and collaborative effort among students. It values benchmarking and the selection of competitive models, recognizing their unique features across various domains, and devising strategies to close the performance gap, thereby aspiring to reach the level of these exemplary competitors. Thus, the importance of competitiveness in our university lies in maximizing the use of all available resources within educational institutions to achieve the best outcomes that align with international quality standards and the needs of the labor market [4,8].

The thematic analysis results (Table 2) indicate that KKU's competitive strengths, as identified by participants, encompass a comprehensive quality management system, mentioned 24 times, and a dedication to ongoing improvement to satisfy customer requirements, cited 9 times. Participant 3 noted, "*KKU is making significant strides and sincere efforts towards development and modernization by placing quality concepts at the forefront of its developmental strategy, which serves as the foundational framework for all its administrative and academic operations*." Moreover, participant 6 remarked, "*KKU, with its extensive quality system and diverse roles, has distinguished itself as an educational institution in a prime geographical location, achieving high efficiency and meeting the needs of individuals in a rapidly evolving era, thus maintaining its competitive edge among local, regional, and international higher education institutions*."

The organizational climate at KKU is characterized by a strong sense of positivity, as noted by 9 participants. The first participant mentioned, "*Love and positivity are plentiful among the university's staff and leaders*," while Participant 8 commented, "*The work environment at the university is positive, motivating, and supportive for all staff members*." Furthermore, organizational trust is a prominent aspect of KKU's work environment, mentioned six times. Participant 4 stated, "*Mutual trust and respect are prevalent across all colleges and units of

*the university*," and Participant 11 observed, "***Trust between leaders and employees is clear, promoting good working relationships marked by cooperation.***"

King Khalid University actively promotes its offerings both domestically and internationally. The institution supports business incubators and maintains collaborations with entities in the public and private sectors, as highlighted by twelve participants. Participant 3 reported, "***The university has provided business incubators for students and faculty members.***" Participant 6 added, "***In recent years, the university has engaged in partnerships with various donors from the public and private sectors within the Kingdom.***" Furthermore, King Khalid University has developed channels to market its products through its marketing unit, as mentioned by nine participants. Participant 14 revealed, "***The university has established a marketing unit under the Agency of Economy and Business to market the university's products.***" Participant 7 emphasized, "***A distinctive feature of King Khalid University is its dedicated unit for marketing its products.***"

The thematic analysis results indicate that the feedback from participants in this study identifies a significant competitive advantage for King Khalid University in its flexible strategic plan, mentioned 18 times by respondents. The university's mission, goals, and competitive initiatives were emphasized 11 times. Participant 5 commented, "***King Khalid University's new 2023 strategy is distinctly competitive, sending a strong message of competitiveness.***" Additionally, Participant 7 noted, "***The university's strategic goals include improving learning outcomes and the quality of the academic environment, among other objectives.***" Furthermore, Participant 11 observed, "***The 2023–2030 strategy of King Khalid University features six competitive initiatives designed to enhance its position in both local and international rankings.***"

Furthermore, seven study participants highlighted the university's strategy as flexible, adapting to real-world demands. Participant 22 remarked, "***In response to the Asir region's development strategy, King Khalid University updated its 2018 to 2023 strategy.***" Participant 25 commented, "***The university's strategy is adaptable, evolving alongside the national trends of Saudi Arabia and its Vision 2030.***" These findings are consistent with Huang & Fei Lee's [27] study, which emphasizes the importance of strategic planning in university management as a key to competitive advantage.

The thematic analysis results show that King Khalid University (KKU) implements a strategic management system that includes a variety of components, such as a wide range of academic specializations. This analysis helps identify the university's strengths and areas for enhancement, thereby enabling it to meet the future demands of the labor market within the Kingdom. The participants mentioned 9 times that KKU offers an assortment of major and minor specializations in its academic programs. For example, Participant 19 mentioned, "***In recent years, King Khalid University has added both major and minor specializations to its curriculum.***" Likewise, Participant 21 noted, "***The variety of specializations across King Khalid University's colleges allows students to select a major and a minor in areas that align with their interests.***"

Moreover, four participants noted that KKU acknowledges its strengths and weaknesses, along with opportunities to counteract threats that might impede its progress and strategic goals. Participant 5 stated, "***King Khalid University systematically conducts self-evaluations of its academic and administrative divisions, and its educational programs, to identify areas in need of enhancement.***" Additionally, Participant 9 remarked, "***King Khalid University regularly reviews its academic curricula to pinpoint strengths and areas for growth, proposing improvements that keep pace with the evolution of academic and professional fields, as well as labor market demands.***"

King Khalid University's focus on future labor market trends was highlighted three times in participants' responses. Participant 6 mentioned, "***King Khalid University performs an annual***

*survey to determine the academic disciplines that will be in demand in the future labor market.*" Participant 21 explained, "*King Khalid University has adapted its academic offerings in various disciplines based on a study to identify the fields that will be needed by the future labor market, thus preparing its graduates with the skills to secure employment and contribute to reducing the unemployment rate.*"

KKU is renowned for its highly professional academic and administrative staff, a sentiment reinforced by analysis results showing this view was shared 15 times in participant feedback. Specifically, six respondents praised KKU's academically distinguished faculty. For instance, Participant 1 remarked, "*A hallmark of KKU is the academic distinction of its faculty in both teaching and scientific research,*" while Participant 24 observed, "*KKU's faculty members boast a strong record of scientific publications in both local and international high-impact journals, alongside their professional teaching excellence.*" These comments align with Al-Sahli's [10] study, which concluded that quality programs and internationally recognized faculty enhance a university's competitive edge.

Additionally, the proficiency of KKU's administrative staff was acknowledged four times. Participant 8 said, "*KKU's administrative staff exhibit professional competence in their roles,*" and Participant 20 observed, "*KKU's employees possess the functional competencies needed to effectively perform their duties, supported by advanced technology.*" The establishment of a human resources management department at KKU was mentioned three times. Participant 10 noted, "*KKU boasts a dedicated human resources management for recruitment, performance evaluation, and staff development,*" while Participant 11 mentioned, "*KKU has transformed the Faculty Affairs Deanship into a human resources management department responsible for the professional appointment and development of both academic and administrative staff.*"

Participants also indicated that KKU offers professional development programs for its academic and administrative staff, a point emphasized twice. Participant 9 observed, "*Resource management provides a variety of training programs to enhance faculty performance,*" while Participant 21 commented, "*Human resources management offers a wide range of training programs for all university staff, including faculty and employees.*" This aligns with Al-Abbad's [11] study, which highlights the importance of developing the teaching profession at universities as a prerequisite for enhancing their competitive capabilities, a goal that cannot be achieved without proper faculty training.

Also, the thematic analysis results reveal that KKU boasts an exceptional technological infrastructure and facilities catering to the needs of its entire staff, as mentioned 14 times. Participants highlighted the university's advanced technological services four times. Participant 5 remarked, "*KKU's buildings and facilities are outstanding, providing high-speed Wi-Fi,*" and Participant 7 noted, "*KKU's facilities are equipped with cutting-edge technology that meets the staff's needs.*" Additionally, it was mentioned three times that KKU's classrooms and laboratories are state-of-the-art and well-equipped. For example, Participant 11 stated, "*The classrooms at KKU are modern, meeting health standards and technological requirements,*" and Participant 19 mentioned, "*The labs are stocked with all necessary materials and tools for students and faculty.*"

The presence of administrative technological communications at KKU was emphasized three times. Participant 21 observed, "*KKU employs an electronic administrative communication system named 'Injaz'.*" Meanwhile, Participant 25 noted, "*KKU facilitates official technical communication between students and faculty through an academic system and Blackboard.*" The availability of electronic courses at KKU was also highlighted by two participants. Participant 8 remarked, "*Undergraduate students at KKU can complete electronic courses via the Blackboard system, along with integrated courses.*" Additionally, Participant

13 pointed out, "***KKU is celebrated for its e-learning initiatives, having received multiple awards, and provides a wide range of electronic courses in various specializations***." The well-equipped central library at KKU was mentioned twice. Participant 17 commented, "***KKU's central library is filled with the latest books, references, and periodicals in various fields***," and Participant 24 observed, "***The central library excels, offering easy access to an extensive collection of books and periodicals in all fields, with clear and published schedules for use***."

King Khalid University's competitive strengths are highlighted by its educational programs, which received recognition 12 times from participants. These programs were specifically praised three times for their advanced nature and alignment with the job market's needs. For example, Participant 8 stated, "***King Khalid University offers diverse study programs with curricula and courses tailored to the job market's demands***." Similarly, Participant 12 noted, "***In 2020, King Khalid University's agency overhauled all study programs, curricula, and courses to meet the job market's requirements and align with academic and professional advancements***." These findings are consistent with Al-Sahli's [10] study, which emphasizes the importance of high-quality programs in enhancing a university's competitive edge. They also agree with Liao & Suprapti's [4] findings, highlighting how the quality of university graduates influences the job market and boosts competitiveness.

The availability of an appropriate educational setting was emphasized three times by participants. Participant 3 stated, "***The learning environment at King Khalid University is conducive for both male and female students, fostering a high-quality university experience***." Participant 11 added, "***King Khalid University offers an engaging and suitable academic atmosphere where students feel psychologically safe and have access to comprehensive facilities and equipment necessary for all pursuits, catering to diverse student populations, including those with disabilities***."

The integration of study plans and courses was underscored by two participants. Participant 9 mentioned, "***The study plans are revised to reflect national trends and employment demands***." Participant 14 observed, "***The university's study plans undergo regular evaluations, with curricula designed to sequence and integrate, equipping graduates with the skills and qualifications the job market seeks***." This aligns with Thomran et al. [28], whose research suggests that a key competitive advantage is graduates possessing knowledge, skills, and values in sync with market needs.

The implementation of flexible and equitable admission policies was also highlighted by 2 participants. Participant 6 commented, "***King Khalid University employs specific and publicly announced electronic systems for student admissions***." Participant 11 noted, "***King Khalid University admits students through a transparent system with conditions that ensure fairness for all applicants.***"

The study's participant responses align with theoretical literature, indicating that competitiveness in higher education institutions depends on several factors: Administrative requirements (1) such as operational efficiency and effectiveness, implementing relevant legislation and policies, developing regulations, plans, and strategies to adapt to changes, forecasting labor market and professional/academic institution demands, conducting feasibility studies and cost-benefit analyses, and incorporating competitiveness principles within the university. It also includes fostering partnerships with the public and private sectors; Knowledge requirements (2) involve updating curricula, enhancing knowledge production and research support, improving academic quality, and aligning academic programs with labor market needs; Human requirements (3) encompass improving faculty academic and research skills, refining recruitment processes to attract top talent, training academic leaders, and promoting excellence and innovation in all aspects of the university; and Organizational requirements (4) such as encouraging flexibility and openness with regional and international universities,

establishing research chairs, and creating centers of expertise. Additionally, the ability to learn from experiences and peers to quickly adapt to changes in the university environment and to refine goals, strategies, policies, and overall administrative practices is essential [3,10,29].

## The challenges faced the KKU in achieving advanced rank in international ratings for the Best Universities

The analysis of participant responses to the second question in the study, as shown in Table 3, reveals their perspectives on the challenges KKU faces in climbing the ranks of international university standings. Academic hurdles are notably significant, mentioned 23 times by participants regarding this question. In particular, the classification and accreditation of academic programs were emphasized 10 times. Participant 5 noted, "***KKU's advancement in international university rankings is hindered by the national and international classification of its programs by academic accreditation bodies." Similarly, Participant 25 stated, "The main obstacle to KKU's ascent in international rankings is the accreditation of*** its academic programs." These findings align with Al-Humaidi's [29] research, which highlights the necessity of aligning the university's programs and colleges with quality standards and academic accreditation to secure a competitive edge.

Furthermore, seven participants highlighted that the categorization of KKU's graduates by professional and specialized entities presents a significant barrier to elevating its position among the world's leading universities. Participant 9 stated, "***Graduates from certain disciplines find it difficult to gain recognition from specialized entities.***" Similarly, Participant 12 remarked, "***Graduates of certain programs at KKU encounter obstacles in obtaining professional categorization.***"

In addition to the academic challenges KKU faces in its pursuit to rank as a top-tier global institution, the rate of scientific publication has been underscored six times. Participant 3 noted, "***There is a marked decline in the quantity of international scientific publications by KKU's faculty and researchers in fields like humanities and education, posing a major impediment to the university's goal of ascending the international rankings.***" Moreover, Participant 18 commented, "***The rate of international publications in journals indexed by Clarivate Analytics and Scopus is inadequate and below the threshold required for KKU to advance in global rankings soon.***" These findings are consistent with Rogach et al.'s [30] research, which identified research publication and its quality as crucial elements in competitive standing.

Also, the thematic analysis of the study's participant data indicates that challenges related to human resources, especially concerning faculty and staff, significantly hinder KKU's advancement in both local and international rankings. This issue was highlighted 20 times. The difficulty in recruitment and employment was noted by nine participants. For instance, Participant 11 mentioned, "***The university has struggled for years with the recruitment and hiring of faculty, transitioning to temporary collaborative contracts rather than permanent positions.***" Similarly, Participant 15 observed, "***KKU has ceased offering permanent staff positions, opting instead for temporary contract systems.***" The resistance to change among some educators and administrative staff was mentioned seven times. Participant 9 revealed, "***Several employees are resistant to change and progress,***" while Participant 14 remarked, "***Some faculty members are averse to embracing quality initiatives, development, and change, which can impede the workflow.***"

Moreover, the low faculty-to-student ratio in certain disciplines was emphasized four times. Participant 16 noted, "***Some academic programs are overwhelmed by a surplus of students compared to faculty,***" and Participant 19 commented, "***KKU's specific fields, especially***

*practical ones like health sciences and medicine, suffer from a faculty shortage that affects both male and female students.*"

In the same context, the thematic analysis of responses to the second question revealed that KKU faces significant administrative challenges, mentioned 18 times. A prominent concern is bureaucratic red tape, highlighted 12 times. Participant 5 stated, "***Bureaucracy hampers progress, as evolution and change require decentralized decision-making,***" and Participant 22 observed, "***Centralized decision-making at institutions, including KKU, greatly hinders development.***" The lack of financial resources for certain colleges and programs, affecting their performance in international rankings, was noted six times. Participant 11 mentioned, "***Specific colleges and programs, such as engineering, suffer from insufficient financial support, limiting their improvement and accreditation,***" while Participant 23 pointed out, "***The main barrier to accrediting programs is the financial investment and the provision of necessary teaching equipment.***" These findings align with Al-Sahli's [10] study, which indicated that the university's reputation is undermined by the lack of adequate material and motivational incentives for researchers. Zamir et al. [31] reported that educational funding influences human capital development. As well as Abbasi et al. [32] concluded that the education and health budgets slightly alleviate environmental degradation.

The results of this question also coincide with Al-Abbad's [11] study, which identified several challenges hindering Saudi universities from enhancing their competitive capabilities. Hashem's [3] study highlighted the importance of achieving excellence in learning and teaching, scientific research, community service, and developing plans to improve operations through a strategic vision and goals focused on excellence and competition.

The escalating challenge of competitiveness in higher education institutions necessitates a comprehensive review of their organizational and academic positions. Universities are required to optimize their resources, restructure their units, streamline their operations, and leverage all available educational tools and capabilities. This preparation enables higher education institutions to equip communities with the necessary competencies, skills, and knowledge to enhance and increase their competitive edge. Such initiatives align with the pressures of international competition and the quest for excellence across various areas, including academic programs, faculty, libraries, lecture halls, and research facilities. The goal is to create an environment that promotes productivity and output, efficiently manages resources, and outperforms both local and international competitors. This strategy acknowledges that human capital is a crucial factor in boosting a nation's competitiveness [7,19,33].

As highlighted by [4,5,8,19] the competitiveness of higher education institutions depends on several factors: natural and physical resources, nationally or internationally accredited academic programs, strategic management, and human resources represented by skilled administrative staff. Faculty members with academic qualifications from prestigious universities and the scientific capability to excel in scholarly and research activities are essential. Additionally, a robust infrastructure that encompasses lecture halls, equipment, and superior learning materials to satisfy the needs of all stakeholders, along with fast, reliable, secure, and confidential technological and information systems, is key to achieving competitiveness through quality, innovation, and creativity.

## Strategies to develop KKU's competitiveness capabilities

Table 4 presents the results of the thematic analysis of participants' views on King Khalid University's (KKU) advancement in competitive capacity relative to other universities, both locally and globally. This analysis highlights the essential need for KKU to implement competitive strategies in academia, a viewpoint reiterated by the study's participants 28 times. Eight

participants emphasized the critical need to align academic programs with the demands of the labor market. Participant 1 noted, "***Despite ongoing updates, some programs at KKU still need realignment with the labor market***," while Participant 8 stated, "***Humanities programs at KKU should bridge the gap between their curriculum and the labor market's demand for such fields, identifying disciplines with future market relevance***."

Additionally, the suggestion to introduce new programs was made 5 times in response to the third question. Participant 9 commented, "***The labor market has recently been in search of new specialties and updated curricula***," and Participant 13 highlighted, "***The labor market has moved away from traditional fields towards future-focused specialties and innovative programs***." The idea of attracting top-tier students was mentioned 4 times. Participant 18 suggested, "***To improve first-year student retention rates, KKU should launch a program to attract and support high-achieving students***," and Participant 27 argued, "***The university needs to attract exceptional students who not only graduate on time but also help in winning local and international awards***."

The establishment of specialized centers was underscored three times in the study responses. Participant 27 noted, "***The university lacks specialized centers that contribute to solving the environmental issues of the Asir region***," while Participant 29 suggested, "***The university needs to expand the creation of community-beneficial centers, such as those for environmental research, and research on children and the elderly***."

Similarly, the need to enhance funding for academic programs and scientific research was also mentioned three times. Participant 11 observed, "***Academic programs require financial allocations to provide the necessary equipment and tools for student education***," and Participant 17 emphasized, "***There is a need for financial support and equipment provision to boost the rate of scientific publication in various disciplines***."

Likewise, the importance of strengthening scientific and academic connections between KKU and other prominent institutions was highlighted twice. Participant 9 stated, "***Student exchange programs with international universities provide KKU a competitive capability***," while Participant 24 remarked, "***Faculty members engaging in scientific exchanges and research at esteemed universities will boost their productivity and elevate KKU's competitive stance***." These findings are consistent with Hashim's [3] study, which underscored the imperative for universities to forge collaborations with various research organizations and centers.

Table 4 presents the outcomes of the thematic analysis regarding responses to the third question, revealing that KKU needs a variety of strategies to boost its human resources competitiveness, a concern mentioned 23 times. The importance of mandatory training for faculty and staff was highlighted 12 times. Participant 4 emphasized, "***Mandatory training in modern teaching and assessment techniques is crucial for both new and existing faculty members***," while Participant 26 noted, "***Comprehensive professional development for all of KKU's human resources, including faculty and staff, is essential for achieving the university's strategic goals***." These findings align with Thomran et al. [28], which argues that professional development is fundamental to increasing competitiveness, and with Hashim's [3] research, underlining the importance of continuous professional growth for academic staff.

The study participants underscored the importance of recruiting distinguished faculty members, with 11 references to its significance. Participant 11 noted, "***The university's academic programs require the hiring of faculty with outstanding academic and research credentials to enhance performance metrics***," while Participant 27 observed, "***Attracting distinguished faculty will improve the educational outcomes of KKU's academic programs***." Additionally, the participants discussed the necessity of competitive administrative strategies to improve KKU's market standing, mentioned 22 times. The need to increase system and procedure flexibility was highlighted 18 times. Participant 16 stated, "***Advancement necessitates***

*adaptable administrative processes and operations across the university's academic and administrative divisions*," and Participant 23 recommended, "*To achieve the university's ambitious strategic goals for a competitive 2030 vision, leadership must transition from centralized control to a more flexible approach in decision-making and operations*."

The significance of enhancing the university's technological infrastructure was underscored by two participants. Participant 26 emphasized, "*To realize the university's ambitious vision, we must globalize education, not confine it to regional limits, necessitating a comprehensive upgrade of the technological infrastructure in all academic activities at KKU*." In a similar vein, Participant 30 noted, "*A crucial aspect of competitiveness is the advancement of KKU's technological infrastructure in both academic and administrative areas to bolster development initiatives*." These insights align with the findings of Ekeagbara et al. [34], which highlight the essential need for universities to adopt competitive administrative strategies.

These findings align with the emphasis by Ghemawat & Rivkin [17] on the necessity for universities to devise strategies and practical guides for continuously enhancing their competitiveness. This involves cultivating a competitive culture within the university community, integrating it into development plans, forming committees to identify, monitor, and assess competitive practices, and generating regular reports. Universities should set performance benchmarks aimed at achieving both local and global competitiveness, ensure cohesive integration and coordination across all administrative and academic sectors to boost competitiveness, and steer the educational process towards developing skilled, creative, and practical competencies to produce graduates who are highly competitive on both local and global scales. Providing the university with sufficient infrastructure, technology, communication systems, learning resources, and educational and research tools designed to meet academic, administrative, and research requirements is vital for attaining competitive capabilities. Furthermore, implementing effective governance, enhancing institutional effectiveness, improving the work environment, minimizing bureaucracy, developing programs for modern and future sciences, adopting advanced educational systems, strengthening connections between educational institutions, promoting research collaboration and excellence, nurturing talent, and establishing awards for exceptional university performance are all critical components.

Also, Grant [15], Huang & Fei Lee [27], and Ehmke [35] have highlighted the significance of enhancing universities' competitive edge by fostering information and communication technology, student engagement, community service effectiveness, and environmental initiatives. It is essential to strongly focus on their educational mission while incorporating feedback from society. Integrating an international perspective into university strategies and visions is crucial for delivering state-of-the-art, competitive higher education that contributes to a knowledge-based society and meets the needs of economic, social, and environmental advancement. Consequently, Al-Baghdadi [14] emphasized the importance of identifying the skills that university graduates require to compete on local, regional, or international levels, securing significant international publications in prestigious journals, and strengthening partnerships with academic institutions and research organizations through faculty exchanges, student scholarships, and similar activities to boost universities' competitive positions.

Moreover, what sets one university apart from another in terms of competitive capability should originate from the institution itself, turning its unique advantage into a competitive strength. Competitive advantage is developed, not inherited, and is driven by competitive creativity, with innovation being the cornerstone of this advantage, rather than factors that occur naturally. It requires a transition from static to technological inputs, with innovation serving as a local, not international, input. Strategies that allow a university to develop a competitive strategy include technology, strategic management, and total quality management as ways to achieve competitiveness [36,37].

## Conclusion

The current study aimed to understand the perspectives of participants regarding KKU's competitive strengths. It sought to identify essential strategies to boost its competitiveness and rise in global university rankings. Utilizing a qualitative approach, the study conducted a case analysis of KKU. The results showed that the university possesses several competitive advantages. However, the study also revealed that KKU faces multiple challenges that could hinder its pursuit of higher positions in global university rankings. These challenges encompass academic issues, human resources, and administrative obstacles. Also, participants in the study suggested several strategies to improve KKU's competitive edge, focusing on academic, human, and administrative competitiveness mechanisms.

## Limitations and future directions

While this study offers valuable insights, it is not devoid of limitations, such as its qualitative nature and the sole focus on KKU. It also did not utilize a mixed-methods approach, which could have enhanced the findings and was confined to a single institution. Future research should aim to bolster the competitive capabilities of higher education institutions by adopting diverse methodologies, including mixed methods, and by exploring a wider array of universities. Given these findings, subsequent studies should explore the competitive strengths of other Saudi universities to fully grasp their potential to compete on both local and international levels. Such research should identify what these institutions require to boost their competitiveness, aiming to place at least five Saudi universities in the top 100 or 200 of international rankings by 2030, in line with the national vision. The outcomes of this study highlight the necessity to reassess the university education system to address new challenges and meet societal needs, thereby enhancing competitiveness in the knowledge economy and internationally, and ultimately fulfilling the national goal of establishing world-class universities.

## Acknowledgments

The authors would like to express their gratitude to King Khalid University, Saudi Arabia, for providing administrative and technical support.

## Author Contributions

**Conceptualization:** Boshra Ismael Ahmed Arnout, Thabet Saeed AlQahtani, Hessah A. L. Melweth.

**Data curation:** Boshra Ismael Ahmed Arnout.

**Formal analysis:** Boshra Ismael Ahmed Arnout, Thabet Saeed AlQahtani, Hessah A. L. Melweth.

**Investigation:** Thabet Saeed AlQahtani, Hessah A. L. Melweth.

**Methodology:** Boshra Ismael Ahmed Arnout, Thabet Saeed AlQahtani, Hessah A. L. Melweth.

**Resources:** Boshra Ismael Ahmed Arnout, Hessah A. L. Melweth.

**Supervision:** Thabet Saeed AlQahtani.

**Validation:** Boshra Ismael Ahmed Arnout.

**Writing – original draft:** Boshra Ismael Ahmed Arnout, Thabet Saeed AlQahtani.

**Writing – review & editing:** Boshra Ismael Ahmed Arnout, Thabet Saeed AlQahtani.

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
