## [Decision Letter · Decision Letter 0]

15 Feb 2024

PONE-D-23-42964Competitive Capabilities of Higher Education Institutions: A Case Study of King Khalid University (KKU)PLOS ONE

Dear Dr. Arnout,

Thank you for submitting your manuscript to PLOS ONE. After careful consideration, we feel that it has merit but does not fully meet PLOS ONE’s publication criteria as it currently stands. Therefore, we invite you to submit a revised version of the manuscript that addresses the points raised during the review process.

We look forward to receiving your revised manuscript.

Kind regards,

Yasir Ahmad

Academic Editor

PLOS ONE

Journal Requirements:

2. You indicated that ethical approval was not necessary for your study. We understand that the framework for ethical oversight requirements for studies of this type may differ depending on the setting and we would appreciate some further clarification regarding your research. Could you please provide further details on why your study is exempt from the need for approval and confirmation from your institutional review board or research ethics committee (e.g., in the form of a letter or email correspondence) that ethics review was not necessary for this study? Please include a copy of the correspondence as an ""Other"" file.

4. We note that you have referenced (Unpublished) on page 16, which has currently not yet been accepted for publication. Please remove this from your References and amend this to state in the body of your manuscript: (ie “Bewick et al. [Unpublished]”) as detailed online in our guide for authors

5. Please amend your manuscript to include your abstract after the title page.

Reviewers' comments:

Reviewer's Responses to Questions

**Comments to the Author**

1. Is the manuscript technically sound, and do the data support the conclusions?

Reviewer #1: Yes

Reviewer #2: Partly

Reviewer #3: Yes

Reviewer #4: Partly

2. Has the statistical analysis been performed appropriately and rigorously? 

Reviewer #1: Yes

Reviewer #2: Yes

Reviewer #3: Yes

Reviewer #4: N/A

3. Have the authors made all data underlying the findings in their manuscript fully available?

Reviewer #1: Yes

Reviewer #2: Yes

Reviewer #3: Yes

Reviewer #4: No

4. Is the manuscript presented in an intelligible fashion and written in standard English?

Reviewer #1: Yes

Reviewer #2: Yes

Reviewer #3: No

Reviewer #4: No

5. Review Comments to the Author

Reviewer #1: This study utilizes a qualitative case study methodology to uncover King Khalid University's (KKU) 30 sub-competitive strengths, including academic prowess, collaboration, and technological services, among others. Thematic analysis of data from in-depth interviews with university personnel highlights 8 challenges hindering KKU's international ranking ascent and proposes 11 strategies to bolster its competitive capabilities. These findings underscore the importance of academic, human, and administrative strategies for KKU to achieve its goals and align with Saudi Arabia's National Vision 2030 of elevating several universities to global prominence.

• The abstract is too long.

• The use of references is outdated. Authors should utilize updated references that are not older than five years.

• Introduction should be updated. Literature review should be added. At present, I cannot see the review. To enhance your introduction and literature review, consider citing the following papers:

• https://doi.org/10.1016/j.heliyon.2023.e20417

• https://doi.org/10.3390/su142114586

• https://doi.org/10.1007/s43545-023-00711-w

• https://doi.org/10.1007/s13132-023-01115-0

Reviewer #2: References in the text should be clarified. For example (Supe & Inguna), Inguna is the First name, it should be written (Supe & Jurgelane), also in 17pp Al-Abbad 2017. would recommend Al-Abbad (2017), please check all references.

In the Participants section, you could indicate the total number of university teaching staff, this would create more confidence in the accuracy of the data

You might reconsider the usefulness of Figures, since they basically duplicate information from the Tables

Reviewer #3: The authors adhered to all the questions above. The data perfectly support the conclusions.

The statistical analysis were done appropriately with a modern software for qualitative data analysis.

All the findings in the manuscript are available.

There are a few grammatical errors that the authors should edit and some sentences also need to be broken down.

Reviewer #4: This study seems interesting. However, there are several methodological and research issues to be addressed seriously. Below are some of the many problems found.

1. The title and research do not reflect novelty.

2. How many strengths are listed in the abstract? Why do you need to mention all of them? It is very long.

3. How do you define academic freedom, intellectual property rights, and the two experts in your context?

4. The research gap is not spelled out clearly nor sufficiently.

5. The first RQ looks like an interview question with the use of personal pronoun ‘your’. The RQs are listed under the objectives; this is confusing.

6. Only three interview questions are mentioned in the first paragraph of the Method section, let alone how they are developed and validated. This is invalid.

7. The demographic data and section 3.3. Data Analysis are very brief. Is the interview individual or group? Why so?

8. No information is given about MAXQDA 2022 software. Why using this program, not others?

9. No citations in sections 3.3 and 3.4.

10. What do you mean by open and closed codes? Why needing Table 1 and Fig. 1 & 2 at the same time? This applies to the others below them.

11. The discussion section is discursively written with more than 6 pages without any sub-headings to function as signposts to guide the reader.

12. The conclusion section is around 12 lines, but the abstract is around 22 lines. Is that reasonable?

13. There is no theoretical framework/section to guide the study.

14. Thence, the methodology is flawed broadly.

15. Overall, the article is disorganised, and poorly written without academic rigour.

16. Please proofread/edit the manuscript to resolve language issues. There are some language issues such as:

a. This study aimed to revealed the KKU's …….

b. Department of Learning and Instructure

c. Limitations Future Directions:

6. PLOS authors have the option to publish the peer review history of their article (what does this mean?). If published, this will include your full peer review and any attached files.

Reviewer #1: No

Reviewer #2: No

Reviewer #3: No

Reviewer #4: **Yes: **Ghayth Al-Shaibani

---

## [Author Response · Author response to Decision Letter 0]

7 Mar 2024

Title: Competitive Capabilities of Higher Education Institutions: A Case Study of King Khalid University (KKU)

The title of the abstract is specific to a certain institution making it more localized. Besides, two authors affiliated with the university where the research took place can affect the results.

Reply: 

Thank you. The current study seeks to explore the competitive capabilities of King Khalid University through the lens of its employees. Consequently, in response to your feedback, we have revised the study's title to: "Competitive Capabilities of Higher Education Institutions from Their Employees' Perspectives: A Case Study of King Khalid University". This research topic is open for investigation by scholars from the same institution. Moreover, this study employs qualitative research methods, as highlighted by Creswell, to ensure its credibility and reliability. These methods include the use of a triangulation strategy in both researcher perspectives and data collection techniques. Additionally, the data analysis was conducted by an expert in qualitative data analysis.

Abstract:

The KKU first mentioned in the abstract should be written in full.

Reply: 

Thank you, we changed it

The first sentence in the abstract KKU’s competitive strength in what?

Reply: 

Thank you, we added (as a higher educational institute) 

The fourth sentence on the results revealed that ………………… is too long and should be restructured.

Reply: 

Thank you, we restructured it. 

Thematic analysis results revealed that……….. the use of thematic analysis was becoming more common in the abstract.

Reply: 

Thank you, the thematic analysis changed to ((the data were qualitatively analyzed using MAXQDA 2022 software)).

Introduction

After the introduction, the sub-headings that came after should be well-known to the audience as to whether they are literature reviews or not. Again, I was expecting that you say something about KKUs in terms of the institutions' competitiveness with other institutions and the need for this study briefly. This will inform readers what is expected to follow in subsequent sections of the work. 

Reply: 

Thank you, we changed the title of the introduction section to ((Introduction and Theoretical Background))

Methodology

Method

The first sentence is it depth interviews or in-depth interviews?

Reply: 

Thank you, we changed it.

Participants

How do you ensure that there were no biases in the study as two of the participants were from the university?

Reply: 

Given the objective of this study to examine King Khalid University's competitive edge through the lens of its employees and faculty members, the sample must comprise individuals employed at King Khalid University. These individuals are acquainted with the institution's deficiencies, weaknesses, and the obstacles they encounter in their roles. Should the sample include individuals external to King Khalid University, they would lack insight into these issues. Nonetheless, an investigation into King Khalid University's competitive advantage could also be conducted from the perspective of decision-makers in the Ministry of Education or from the viewpoint of community members.

Discussion

The first paragraph of the discussion should state the key findings of the study. This should be followed by subsequent sections trying to link the findings with previous literature. 

Reply: 

Thank you, we changed it. 

Thank you for your efforts in reviewing the manuscript, as it has helped us improve its quality.

Reviewer #1: 

This study utilizes a qualitative case study methodology to uncover King Khalid University's (KKU) 30 sub-competitive strengths, including academic prowess, collaboration, and technological services, among others. Thematic analysis of data from in-depth interviews with university personnel highlights 8 challenges hindering KKU's international ranking ascent and proposes 11 strategies to bolster its competitive capabilities. These findings underscore the importance of academic, human, and administrative strategies for KKU to achieve its goals and align with Saudi Arabia's National Vision 2030 of elevating several universities to global prominence.

• The abstract is too long.

• The use of references is outdated. Authors should utilize updated references that are not older than five years.

• Introduction should be updated. Literature review should be added. At present, I cannot see the review. To enhance your introduction and literature review, consider citing the following papers:

• https://doi.org/10.1016/j.heliyon.2023.e20417

• https://doi.org/10.3390/su142114586

• https://doi.org/10.1007/s43545-023-00711-w

• https://doi.org/10.1007/s13132-023-01115-0

Reply:

Thank you for these esteemed references, I cited them in the manuscript.

Reviewer #2: 

References in the text should be clarified. For example (Supe & Inguna), Inguna is the First name, it should be written (Supe & Jurgelane), also in 17pp Al-Abbad 2017. would recommend Al-Abbad (2017), please check all references.

Reply: 

Thank you, we edited and changed all of them. 

In the Participants section, you could indicate the total number of university teaching staff, this would create more confidence in the accuracy of the data

Reply: 

Thank you, King Khalid University boasts a total of 3,407 faculty members and 684 administrative employees. A purposive sample of academic leaders and faculty members at King Khalid University was selected to gather detailed insights into KKU's competitive strengths.

Because our study applied the qualitative research design, we chose 30 participants only from the faculty staff members and academic leaders and utilized an in-depth interview tool to collect data from them.

You might reconsider the usefulness of Figures, since they basically duplicate information from the Tables

Reply: 

Thank you, we replaced it with the code-subcode-segment model of participants' responses to the three questions via MAXQDA Software.

Reviewer #3: 

The authors adhered to all the questions above. The data perfectly support the conclusions.

The statistical analysis were done appropriately with a modern software for qualitative data analysis.

All the findings in the manuscript are available.

There are a few grammatical errors that the authors should edit and some sentences also need to be broken down.

Reply: 

Thank you, we reviewed all the manuscript readability and edited all the grammar errors. 

Reviewer #4: 

This study seems interesting. However, there are several methodological and research issues to be addressed seriously. Below are some of the many problems found.

1. The title and research do not reflect novelty.

Reply:

In response to your feedback, we have revised the study's title to: "Competitive Capabilities of Higher Education Institutions from Their Employees' Perspectives: A Case Study of King Khalid University". Also, this study is one of the first to explore King Khalid University's competitive advantage from the perspective of its employees, uncovering the strategies necessary for the university to secure a prominent spot in the global top 100 university rankings, to help King Khalid University achieve the 2030 nation view. 

2. How many strengths are listed in the abstract? Why do you need to mention all of them? It is very long.

Reply:

Thank you, we summarized it, and deleted more of them. 

3. How do you define academic freedom, intellectual property rights, and the two experts in your context?

Reply:

Thank you, At King Khalid University, specific regulations govern the design of academic programs and courses, permitting academic colleges and departments to adjust them by up to 20%. This exemplifies the academic freedom afforded by the university. In terms of intellectual property rights, King Khalid University boasts an Intellectual Property Unit and a Research Ethics Committee. These bodies are tasked with ensuring that researchers at the university adhere to the principles of scientific research ethics and intellectual property rights.

A prestigious university consistently ranks within the top 100 or 200 in international rankings, relies on a single funding source, and enjoys the highest level of academic freedom to develop its programs and conduct training and research in an environment that protects intellectual property rights. In this era, competitiveness presents a significant challenge for universities, forcing them to reevaluate their organizational structures and production capacities, reorganize their units, and optimize their resources to improve and sustain their competitive position and sustainability (Diab, 2014; Satsyk, 2014; Supe et al., 2018).

4. The research gap is not spelled out clearly nor sufficiently.

Reply:

Thank you, we edited and highlighted it on page 4. 

5. The first RQ looks like an interview question with the use of personal pronoun ‘your’. The RQs are listed under the objectives; this is confusing.

Reply:

Thank you, we changed it.

6. Only three interview questions are mentioned in the first paragraph of the Method section, let alone how they are developed and validated. This is invalid.

Reply:

Thank you, according to the objectives and questions of the current study, we formulated many open-ended questions to present during the participants' interview, such as: What are your views on KKU's competitive advantages? What challenges does KKU face in its pursuit of higher rankings in international university standings? How can KKU enhance its global competitiveness? What strategies are essential for achieving this objective? and other. 

7. The demographic data and section 3.3. Data Analysis are very brief. Is the interview individual or group? Why so?

Reply:

Thank you, we added it to the method section

8. No information is given about MAXQDA 2022 software. Why use this program, not others?

Reply:

Thank you, MAXQDA is a software program for the analysis of qualitative data, and we use it because we have excellent experts in using it. We used it in many of our studies and trained the Arab researchers to use it. It is not suitable to write it in the manuscript. 

9. No citations in sections 3.3 and 3.4.

Reply:

Thank you, we edited it. 

10. What do you mean by open and closed codes? Why needing Table 1 and Fig. 1 & 2 at the same time? This applies to the others below them.

Reply: 

Thank you, we replaced it with the code-subcode-segment model of participants' responses to the three questions via MAXQDA Software.

11. The discussion section is discursively written with more than 6 pages without any sub-headings to function as signposts to guide the reader.

Reply: 

Thank you, we edited it.

12. The conclusion section is around 12 lines, but the abstract is around 22 lines. Is that reasonable?

Reply: 

Thank you, we edited the abstract. 

13. There is no theoretical framework/section to guide the study.

Reply: 

Thank you, the theoretical background we have written with the introduction section, thus we changed the name of this section to (Introduction and Theoretical Background) (Please, see pages 1-4). 

14. Thence, the methodology is flawed broadly.

Reply: 

Thank you, we edited it (please, see page 5)

15. Overall, the article is disorganised, and poorly written without academic rigour.

Reply: 

Thank you, we edited our manuscript according to PlOSONE, and in our qualitative study, the academic rigor is verified through credibility and transferability, please read the following on page 6: 

3.1. Credibility, Reliability, and Transferability:

Credibility in this study was established through participants' consistent responses. Researchers implemented various strategies to ensure data credibility, including meticulous re-reading and transcription of raw data from participant interviews, and individual analysis of each interview, followed by a comprehensive analysis to synthesize the data. Additionally, ensuring saturation was confirmed by selecting suitable participants. Reliability was ascertained through triangulation and extensive data collection from individual interviews by several researchers, along with independent coding of data by two experts in qualitative thematic analysis. Regarding transferability, extensive data were compiled to allow readers to evaluate the applicability of the study's findings to other contexts (Maxwell,2012, Arnout, 2020).

16. Please proofread/edit the manuscript to resolve language issues. There are some language issues such as:

a. This study aimed to revealed the KKU's …….

b. Department of Learning and Instructure

c. Limitations Future Directions:

Reply: 

Thank you, we have reviewed the readability of our manuscript and proofread it.

Please note that (Department of Learning and Instructure) is correct, this is one of the departments of the education college at King Khalid University.

---

## [Decision Letter · Decision Letter 1]

16 Apr 2024

Competitive Capabilities of Higher Education Institutions from Their Employees' Perspectives: A Case Study of King Khalid University

PONE-D-23-42964R1

Dear Dr. Ahmed,

We’re pleased to inform you that your manuscript has been judged scientifically suitable for publication and will be formally accepted for publication once it meets all outstanding technical requirements.

Kind regards,

Yasir Ahmad

Academic Editor

PLOS ONE

Additional Editor Comments (optional):

Reviewers' comments:

Reviewer's Responses to Questions

**Comments to the Author**

1. If the authors have adequately addressed your comments raised in a previous round of review and you feel that this manuscript is now acceptable for publication, you may indicate that here to bypass the “Comments to the Author” section, enter your conflict of interest statement in the “Confidential to Editor” section, and submit your "Accept" recommendation.

Reviewer #1: All comments have been addressed

Reviewer #2: All comments have been addressed

2. Is the manuscript technically sound, and do the data support the conclusions?

Reviewer #1: Yes

Reviewer #2: Yes

3. Has the statistical analysis been performed appropriately and rigorously? 

Reviewer #1: Yes

Reviewer #2: Yes

4. Have the authors made all data underlying the findings in their manuscript fully available?

Reviewer #1: Yes

Reviewer #2: Yes

5. Is the manuscript presented in an intelligible fashion and written in standard English?

Reviewer #1: Yes

Reviewer #2: Yes

6. Review Comments to the Author

Reviewer #1: The paper is complete and accepted now in its present form.

Reviewer #2: (No Response)

7. PLOS authors have the option to publish the peer review history of their article (what does this mean?). If published, this will include your full peer review and any attached files.

Reviewer #1: No

Reviewer #2: No
